# Reduced Tie2 in Microvascular Endothelial Cells Is Associated with Organ-Specific Adhesion Molecule Expression in Murine Health and Endotoxemia

**DOI:** 10.3390/cells12141850

**Published:** 2023-07-14

**Authors:** Peter J. Zwiers, Jacqueline P. F. E. Lucas, Rianne M. Jongman, Matijs van Meurs, Eliane R. Popa, Grietje Molema

**Affiliations:** 1Department of Pathology and Medical Biology, Medical Biology Section, University Medical Center Groningen, University of Groningen, P.O. Box 30.001, 9700 RB Groningen, The Netherlands; j.p.f.e.lucas@umcg.nl (J.P.F.E.L.);; 2Department of Critical Care, University Medical Center Groningen, University of Groningen, P.O. Box 30.001, 9700 RB Groningen, The Netherlands; 3Department of Anesthesiology, University Medical Center Groningen, University of Groningen, P.O. Box 30.001, 9700 RB Groningen, The Netherlands

**Keywords:** adhesion molecules, endothelial cells, endothelial heterogeneity, endotoxemia, E-selectin, inflammation, kidney, lung, microvasculature, Tie2 knockout mouse model, VCAM-1

## Abstract

Endothelial cells (ECs) in the microvasculature in organs are active participants in the pathophysiology of sepsis. Tyrosine protein kinase receptor Tie2 (Tek; Tunica interna Endothelial cell Kinase) is thought to play a role in their inflammatory response, yet data are inconclusive. We investigated acute endotoxemia-induced changes in the expression of Tie2 and inflammation-associated endothelial adhesion molecules E-selectin and VCAM-1 (vascular cell adhesion molecule-1) in kidneys and lungs in inducible, EC-specific Tie2 knockout mice. The extent of Tie2 knockout in healthy mice differed between microvascular beds, with low to absent expression in arterioles in kidneys and in capillaries in lungs. In kidneys, *Tie2* mRNA dropped more than 70% upon challenge with lipopolysaccharide (LPS) in both genotypes, with no change in protein. In renal arterioles, tamoxifen-induced Tie2 knockout was associated with higher VCAM-1 protein expression in healthy conditions. This did not increase further upon challenge of mice with LPS, in contrast to the increased expression occurring in control mice. Also, in lungs, *Tie2* mRNA levels dropped within 4 h after LPS challenge in both genotypes, while Tie2 protein levels did not change. In alveolar capillaries, where tamoxifen-induced Tie2 knockout did not affect the basal expression of either adhesion molecule, a 4-fold higher E-selectin protein expression was observed after exposure to LPS compared to controls. The here-revealed heterogeneous effects of absence of Tie2 in ECs in kidney and lung microvasculature in health and in response to acute inflammatory activation calls for further in vivo investigations into the role of Tie2 in EC behavior.

## 1. Introduction

Sepsis is a life-threatening condition with extensive physiological and biochemical abnormalities, and is defined as organ dysfunction caused by a dysregulated host response to an infection [1]. Approximately 49 million people are affected every year by sepsis, accounting for almost 20% of all deaths globally [2]. Acute kidney injury (AKI) and acute respiratory distress syndrome (ARDS) are particularly common complications of sepsis, yet are still poorly understood molecularly. There is consensus, however, that an increase in leukocyte recruitment, microvascular permeability, and procoagulant status all contribute to sepsis-related organ dysfunction [3]. The endothelial cells in the microvasculature play an important role in the pathophysiology of sepsis, as they directly engage in these processes [4,5]. Furthermore, their location in the body makes them uniquely equipped to quickly respond to infection-associated systemic changes. Elevated plasma levels of soluble adhesion molecules and other (micro)vasculature-associated proteins in patients with sepsis support the notion that endothelial cells actively participate in sepsis and sepsis-associated multiple organ dysfunction syndrome [6].

Tyrosine protein kinase receptor Tie2 (or Tek, Tunica interna Endothelial cell Kinase) has been implicated in the control of microvascular responses in sepsis and sepsis-related AKI and ARDS. Tie2 has two ligands, of which, in normal conditions, Angiopoietin-1 (Angpt-1) binds and activates the receptor to stabilize the endothelial monolayer [7]. In inflammatory conditions, Angiopoietin-2 (Angpt-2) is released from Weibel–Palade bodies and competes with Angpt-1 for Tie2 binding. This prohibits Tie2 phosphorylation, which leads to endothelial monolayer destabilization and loss of vascular integrity [8]. The Tie2/Angpt system is of interest as a pharmacological target in sepsis, as activation of Tie2 in mice resulted in lower leakage of peritoneal microvasculature induced by polymicrobial sepsis in a cecal ligation and puncture mouse model [9]. Similarly, when Tie2 was activated by an engineered form of Angpt-1, microvascular permeability in the kidney was partly reduced in mice challenged with lipopolysaccharide (LPS) [10].

Molecularly, the Tie2/Angpt system also engages in intracellular signal transduction associated with inflammatory activation of endothelial cells. Tie2 phosphorylation was shown to be associated with inhibition of the NF-κB signal transduction pathway via transcription factor A20 binding inhibitor of NF-κB activation-2 [11]. Additionally, in the presence of Angpt-2, an increased expression of vascular cell adhesion molecule-1 (VCAM-1) and intercellular adhesion molecule-1 (ICAM-1) occurred in TNFα-stimulated HUVEC, and Angpt-1 treatment of HUVEC and mice suppressed upregulation of E-selectin, VCAM-1 and ICAM-1 when exposed to LPS [12,13,14]. At the same time, intervention with Tie2 phosphorylation also exerted an effect on circulating levels of inflammatory cytokines, possibly via Tie2 stabilization in cells of non-endothelial lineage, which may account for the lower inflammatory activation of ECs [9,15]. These studies pointed at a role for Tie2 in the pathophysiology of sepsis, yet the role of endothelially expressed Tie2 in inflammatory activation of ECs remained inconclusive. We therefore developed a transgenic mouse model that enabled specific knockout of Tie2 in ECs. The model employed an inducible knockout system, as transgenic mice lacking Tie2 die in embryo [16]. Moreover, this strategy allowed us to create mice that lack Tie2 in endothelial cells for a short period of time, which prevents long term adaptations of the organism to Tie2 loss. Analysis of Tie2 expression in this *Tie2^ΔE9^* model revealed successful *Tie2* exon9 deletion, with most prominent reduction of Tie2 protein in arterioles in all organs and in the alveolar capillary bed in lungs [17].

The aim of the current study was to investigate whether tamoxifen-induced reduced levels of Tie2 in kidneys and lungs are accompanied by altered expression levels of endothelial cell adhesion molecules in the acute phase of endotoxemia. To achieve this, we challenged *Tie2^∆E9^* knockout and control mice with LPS and studied the effects of this acute inflammatory stimulus on Tie2 expression and on the extent and location of E-selectin and VCAM-1 expression 4 h after the start of endotoxemia. We analyzed mRNA and protein expression levels in kidneys and lungs at whole organ level, and at the level of arterioles in kidneys and capillaries in lungs as these showed the largest extent of Tie2 protein knockout in the *Tie2^∆E9^* mice.

## 2. Materials and Methods

### 2.1. Generation of Tie2^ΔE9^ Knockout Mice and Endotoxemia Model

Endothelial cell-specific knockout of *Tie2* exon 9 in *Tie2^floxed/floxed^*; *end-SCL-Cre-ER^T+/−^* mice was induced via intraperitoneal (i.p.) injection of tamoxifen at a dose of 4 mg/0.1 mL sterile corn oil, three times a week, for a period of three weeks followed by a recovery period of one week (hereafter referred to as *Tie2^ΔE9^*) [17,18]. Littermates that lacked Cre expression were used as controls (*Tie2^floxed/floxed^*; *end-SCL-Cre-ER^T−/−^*, hereafter named *Tie2^fl/fl/Cre-^*) and received tamoxifen according to the regimen described above. Body weights were assessed every 2–3 days during tamoxifen treatment to monitor the welfare of the mice.

At day 28, each genotype group was divided into two subgroups, one of which was i.p. injected with 0.1 mL vehicle (0.9% saline, sterile, B. Braun, Melsungen, Germany), the other i.p. with lipopolysaccharide (LPS; *E. coli*, serotype O26:B6, Sigma-Aldrich, St. Louis, MO, USA, 25 µg/0.1 mL saline) at a dose of 1 µg LPS/g body weight. This resulted in the following four groups of mice: *Tie2^fl/fl/Cre-^*-veh (n = 8; 3 female, 5 male), *Tie2^fl/fl/Cre-^*-LPS (n = 7; 3 female, 4 male), *Tie2^ΔE9^*-veh (n = 7; 2 female, 5 male), and *Tie2^ΔE9^*-LPS (n = 8; 3 female, 5 male). Mice were between 8–28 weeks of age in random distribution over the groups. Four hours after vehicle treatment or endotoxemia induction, mice were sacrificed under anesthesia and organs were harvested, snap frozen on liquid nitrogen, and stored at −80 °C until further processing. The mice in vehicle control groups in this study were previously reported in Zwiers et al. with regard to Tie2 genotype and Tie2 protein levels [17].

Deletion of *Tie2* exon 9 was assessed via genomic DNA PCR using the primers 5′-GGGCTGCTACAATAGCTTTGG-3′ and 5′-GTTATGTCCAGTGTCAATCAC-3′ [19].

### 2.2. RNA Isolation and Quantification of Gene Expression by RT-qPCR

Total RNA was isolated from organs using the RNeasy^®^ Plus Mini Kit and from laser-microdissected kidney material (see below) using the RNeasy^®^ Plus Micro Kit, according to the manufacturer’s protocols (both were obtained from QIAGEN, Venlo, The Netherlands). RNA from laser-microdissected lung alveolar tissue (see below) was isolated using NucleoZOL lysis reagent (Macherey-Nagel, Düren, Germany) with NucleoSpin RNA Set for NucleoZOL (Macherey-Nagel), according to the manufacturer’s protocol. RNA concentration and purity were measured on an ND1000 UV-VIS system (NanoDrop Technologies, Rockland, DE, USA). RNA samples with OD260/280 ratio > 1.9 were included for cDNA synthesis; RNA samples of laser-microdissected tissue were not further analyzed as their concentrations were below the detection limit of standard laboratory equipment. cDNA was synthesized and duplicate quantitative (q)PCRs were performed for each sample using Assay-on-Demand primer/probe sets (Thermo Fisher Scientific, San Diego, CA, USA) as described in Table 1 [17]. The expression of *Cdh5* (also known as and hereafter called *VE-cadherin*) was normalized to the expression of reference gene *Gapdh*. The expression of *Tek* (hereafter called *Tie2)*, and inflammation-induced adhesion molecules *E-selectin (Esel)* and *VCAM-1* were normalized to the expression of *VE-cadherin*. The relative mRNA expression was calculated using the formula 2^−∆Ct^ [20].

### 2.3. Isolation of Renal Microvascular Compartments by Laser Microdissection

To measure mRNA levels of the aforementioned molecules in arterioles in the kidney and capillaries in the lung, these microvascular compartments were isolated via laser microdissection, as described previously [21], prior to the RT-qPCR analysis described above.

### 2.4. Tie2 Protein Quantification by ELISA

Tie2 ELISA was performed as described previously [19]. Amounts of Tie2 protein were normalized to the total protein input of tissue homogenate and expressed as pg/µg of total protein.

### 2.5. Localization of Tie2, E-Selectin, and VCAM-1 Protein by Immunohistochemistry

To determine the location of Tie2, E-selectin, and VCAM-1 proteins in distinct microvascular compartments in mouse kidneys and lungs, immunohistochemical staining was performed as previously described [22]. Incubation with primary monoclonal rat-anti-Tie2 antibody (Tek4, IgG1 isotype; eBioscience, Thermo Fisher Scientific), rat-anti-E-selectin antibody hybridoma supernatant (clone MES-1, IgG2a isotype, a kind gift of Dr Derek Brown, UCB Celltech, Belgium), or rat-anti-VCAM-1 antibody (clone M/K-2, IgG1 isotype, Merck Millipore, Amsterdam, The Netherlands) was followed by mouse absorbed rabbit-anti-rat antibody (Vector Laboratories, Burlingame, CA, USA), and anti-rabbit, horseradish peroxidase-labeled polymer (Dako, Heverlee, Belgium). 3-Amino-9-ethylcarbazole (Sigma-Aldrich, St. Louis, MO, USA) was used for detection, followed by counterstaining with Mayer’s hematoxylin (Merck, Darmstadt, Germany). Isotype control staining with rat IgG1/IgG2a (Antigenix America, New York, NY, USA) was consistently negative (data not shown). Sections of kidneys and lungs (n = 30 mice) were stained in one experiment to avoid inter-experiment staining variability.

### 2.6. Morphometric Quantification of Tie2, E-Selectin, and VCAM-1 Staining in Microvascular Compartments

Stained sections were scanned with a NanoZoomer^®^ 2.0 HT (Hamamatsu Photonics, Almere, The Netherlands) and staining was quantified as previously described, using Aperio Imagescope software v12.2 (Leica Biosystems Imaging, Vista, CA, USA) [19].

### 2.7. Statistical Analysis

Differences in mRNA and protein levels between the four groups were calculated using one-way ANOVA with Sidak correction for multiple comparisons of selected pairs. Differences in vehicle normalized mRNA and protein levels between LPS-challenged *Tie2^fl/fl/Cre-^* control and *Tie2^ΔE9^* knockout mice were calculated using a two-tailed, unpaired *t*-Test. Statistics were performed using GraphPad Prism 9.2.0 (GraphPad Prism Software Inc. La Jolla, CA, USA). Differences were considered statistically significant when *p* < 0.05.

## 3. Results

### 3.1. Effect of Lower Tie2 Levels on Tie2 mRNA and Protein Expression in Endotoxemia in Mice

Previously, we reported that Tie2 knockout in the mouse model employed was variable and microvascular compartment-dependent, with most extensive Tie2 loss occurring in kidneys and lungs (Appendix A) [17]. Here, we first investigated whether tamoxifen-induced reduced levels of Tie2 had an effect on Tie2 expression in response to an inflammatory stimulus, as previously reported [22]. Treatment of the mice with tamoxifen did not affect body weight, nor did we observe altered behavior of the mice. Genomic PCR revealed effective deletion of *Tie2* exon 9 after tamoxifen treatment in both vehicle- and LPS-challenged *Tie2^∆E9^* knockout mice (Appendix A).

Exposure of *Tie2^fl/fl/Cre-^* control mice to LPS caused a 75% reduction in *Tie2* mRNA in kidneys, and a 40% reduction in lungs. The extent of reduction in *Tie2* mRNA was similar in *Tie2^∆E9^* knockout mice (Figure 1A). Since the inducible Tie2 knockout mouse targets the endothelium, and vascular density differs per organ, the pan-endothelial marker gene *VE-cadherin* (*Cdh5*) was used as a reference gene for quantification of *Tie2* mRNA levels. Unpublished results showed, however, that upon LPS challenge *VE-cadherin* expression slightly (~1.6 fold) increased in kidneys of both *Tie2^fl/fl/Cre-^* control mice and *Tie2^∆E9^* knockout mice, while the expression in lungs was unaffected. When using *Gapdh* instead of *VE-cadherin* as a reference gene, the overall result, that LPS induced a rapid downregulation of *Tie2* in both organs, did not change. As correction for endothelial content in samples of organs and laser microdissected microvascular compartments is a prerequisite to compare gene expression data in this study, all mRNA expression analyses were based on reference to *VE-cadherin* mRNA levels.

Protein levels of Tie2 remained unchanged in response to LPS in both organs in both genotypes (Figure 1A). Next, we investigated the extent of the expression of Tie2 protein in the two organs using morphometric analysis of immunohistochemically stained tissue sections (Figure 1B). Tie2 protein quantification of whole kidneys revealed an additional loss of 35% in Tie2 protein in *Tie2^∆E9^* knockout mice compared to an additional 12% loss in *Tie2^fl/fl/Cre-^* control mice in response to LPS (Figure 1C). Although group comparisons did not reveal statistically significant differences in LPS-induced arteriolar Tie2 protein loss between *Tie2^fl/fl/Cre-^* and *Tie2^∆E9^* knockout mice, the data showed that loss occurred in 6 out of 7 *Tie2^∆E9^* mice. No additional loss of Tie2 due to LPS challenge was observed in whole lungs or in lung capillary beds in *Tie2^fl/fl/Cre-^* and *Tie2^∆E9^* knockout mice (Figure 1C).

### 3.2. In Kidneys, Tamoxifen-Induced Reduced Tie2 Expression in Arterioles Did Not Affect E-Selectin and VCAM-1 mRNA Levels in Response to LPS Challenge

We next investigated the mRNA expression levels of inflammation-associated cell adhesion molecules *E-selectin* and *VCAM-1*. *E-selectin* mRNA levels increased in whole kidneys and arterioles of both *Tie2^fl/fl/Cre-^* control (kidney 7.9-fold; arterioles 15.7-fold) and *Tie2^∆E9^* knockout mice (kidney 14.6-fold; arterioles 27.3-fold) after LPS challenge (Figure 2A). This fold increase in kidney was 50% higher in *Tie2^∆E9^* knockout mice compared to control mice, although a large variation in *E-selectin* mRNA levels existed. In arterioles, the microvascular segment with the highest Tie2 knockout in the *Tie2^∆E9^* mice, *E-selectin* also increased in both groups, with fold increase not significantly different between *Tie2^fl/fl/Cre-^* controls and *Tie2^∆E9^* knockout. *VCAM-1* mRNA expression levels also increased after exposure to LPS, both in kidneys as a whole and in arterioles, and this increase was similar for both *Tie2^fl/fl/Cre-^* control (kidney 7.8-fold; arterioles 5.1-fold) and *Tie2^∆E9^* knockout mice (kidney 8.7-fold; arterioles 3.4-fold) (Figure 2B).

### 3.3. In Kidneys, Tamoxifen-Induced Reduced Tie2 Expression in Arterioles Was Associated with Higher Basal VCAM-1 Protein Expression, Yet Did Not Affect LPS Induced E-Selectin and VCAM-1 Protein Expression

Immunohistochemical staining of mouse kidneys showed that, after LPS challenge in both *Tie2^fl/fl/Cre-^* control mice and *Tie2^∆E9^* knockout mice, E-selectin protein was mainly located in the glomerular compartment and in postcapillary venules, and to some extent in arterioles and peritubular capillaries (Figure 3A). Morphometric analyses of the immunohistochemical staining revealed increased E-selectin expression in acute endotoxemia compared to vehicle control (*Tie2^fl/fl/Cre-^* 50-fold; *Tie2^∆E9^* 67-fold), with the extent of induction not being different between both genotypes in kidneys and arterioles (Figure 3B).

VCAM-1 protein was already expressed in control conditions in all renal microvascular beds except glomeruli (Figure 3C). Analysis of the kidney as a whole showed that LPS challenge resulted in a 5.2-fold increase in VCAM-1 protein levels in *Tie2^fl/fl/Cre-^* control mice, while in *Tie2^∆E9^* knockout mice (Figure 3D), the extent of induction compared to control mice was not significant. Zooming in on arterioles revealed a 1.4-fold higher basal arteriolar VCAM-1 protein expression in knockout mice compared to controls (Figure 3D). Arteriolar VCAM-1 levels increased 1.5-fold in response to LPS treatment in *Tie2^fl/fl/Cre-^* control mice, while no change in addition to the higher basal expression level was seen in *Tie2^∆E9^* knockout mice.

### 3.4. In Lungs, Tamoxifen-Induced Reduced Tie2 Expression Did Not Affect E-Selectin or VCAM-1 mRNA Levels in Response to LPS

In lungs of *Tie2^fl/fl/Cre-^* control mice, *E-selectin* mRNA levels increased 4.3-fold in acute endotoxemia, while in *Tie2^∆E9^* knockout mice, induction was less pronounced and statistically not significant compared to vehicle controls (Figure 4A). The extent of induction of *E-selectin* mRNA levels did not differ between genotypes. In alveolar capillaries, induction of *E-selectin* mRNA in response to LPS was absent in both genotypes.

*VCAM-1* mRNA levels in lungs of *Tie2^fl/fl/Cre-^* control and *Tie2^∆E9^* knockout mice increased to a similar extent (2.5-fold and 2-fold, respectively) in response to LPS (Figure 4B). In alveolar capillaries, a 2-fold increase in *VCAM-1* mRNA levels in response to LPS was observed in *Tie2^fl/fl/Cre-^* control mice, while the expression in *Tie2^∆E9^* mice was not significantly elevated. No difference in LPS-induced fold increase in VCAM-1 mRNA levels compared to vehicle controls was seen between *Tie2^fl/fl/Cre-^* control and *Tie2^∆E9^* knockout mice.

### 3.5. In Lungs, Tamoxifen-Induced Reduced Tie2 Expression Was Associated with Higher E-Selectin Protein Levels in Response to LPS

Immunohistological staining of lungs revealed that in vehicle-treated mice, E-selectin protein was hardly present, except for staining of scattered cells in a minority of arterioles. After LPS challenge, E-selectin was mainly located in arterioles and venules, and to a minor extent in alveolar capillaries (Figure 5A). Morphometric quantification of E-selectin in lungs as a whole showed an increased protein expression in both LPS-treated *Tie2^fl/fl/Cre-^* control and *Tie2^∆E9^* knockout mice (2.8-fold and 7.2-fold, respectively; Figure 5B). Furthermore, in *Tie2^∆E9^* knockout mice, E-selectin protein expression in alveolar capillaries in response to LPS administration was 4-fold higher than in *Tie2^fl/fl/Cre-^* control mice.

VCAM-1 protein was expressed in all microvascular segments of lungs of *Tie2^fl/fl/Cre-^* control and *Tie2^∆E9^* knockout mice, both in quiescence and in response to LPS (Figure 5C). Morphometric quantification did not reveal differences in the expression levels between the groups when analyzing the lungs as a whole, nor when focusing specifically on the alveolar capillaries (Figure 5D).

## 4. Discussion

Organ function loss in sepsis is related to excessive leukocyte recruitment and microvascular leakage and a disbalance in anticoagulant/procoagulant status in response to the invading organism. Tie2 and its ligands, Angpt-1 and Angpt-2, play a crucial role in the maintenance of microvascular integrity, while their molecular engagement in the inflammatory activation of endothelial cells in small blood vessels in organs remains elusive. The aim of the current study was to investigate, in an animal model with tamoxifen-induced reduced levels of Tie2 in arterioles in kidneys and capillaries in lungs, whether this lower Tie2 expression was accompanied by an altered expression of Tie2 and adhesion molecules E-selectin and VCAM-1 in the acute phase of endotoxemia. In kidneys and lungs, loss of *Tie2* mRNA occurred in both genotypes upon challenge with LPS, while Tie2 protein levels did not change. In arterioles in the kidneys of 6 out of 7 *Tie2^ΔE9^* knockout mice, an increase in reduction of Tie2 protein in response to LPS was observed compared to the control group. In alveolar capillaries in knockout mice, no additional reduction in Tie2 occurred. Furthermore, in kidney arterioles, tamoxifen-induced reduced Tie2 protein levels were not associated with changes in LPS-induced E-selectin mRNA and protein expression. In contrast, low Tie2 expression in this compartment was associated with higher VCAM-1 protein levels in quiescence, which, upon exposure to LPS, did not further increase. In lungs, tamoxifen-induced reduced levels of Tie2 in the alveolar capillary compartment of knockout mice was associated with 4-fold higher E-selectin protein expression in response to LPS compared to the control genotype, with no effect of Tie2 knockout on VCAM-1. Summarizing, low to absent Tie2 expression in vivo in renal arterioles was associated with higher basal VCAM-1 protein expression, and in lung capillaries with higher endotoxemia-induced E-selectin protein expression.

Our genetic mouse model was based on the SCL-Cre-ERT mouse line that expresses Cre recombinase specifically in the endothelial compartment. In the initial experiment by Göthert et al., crossing with a LacZ reporter mouse demonstrated that Cre recombinase was extensively expressed in endothelial cells in small blood vessels in the majority of organs [18]. However, in our model, the extent of Tie2 knockout was different between endothelial cells in different microvascular beds [17]. By focusing on those microvascular beds that were devoid of Tie2 protein, i.c., arterioles in kidneys and capillaries in lungs, our study revealed heterogeneous consequences of Tie2 absence on inflammatory gene expression in response to LPS. The generation of endothelial-specific Tie2 knockout based on transgenic mice in which Cre recombinase is expressed under the control of, e.g., VE-cadherin as a driver, may reveal whether heterogeneous knockout, as observed in our *Tie2^ΔE9^* model, is a target gene-related limitation or a Cre-driver-related limitation [23]. Furthermore, experiments with such new mouse lines may assist in investigating which mechanism(s) underlie endothelial subset-specific escape from Cre-recombinase-induced deletion of floxed-*Tie2*-exon9. If, in such new models, microvascular loss of Tie2 occurs in all microvascular segments and in multiple organs, studies can be initiated to investigate whether Tie2-controlled inflammatory EC activation is based on (micro)vascular and organotypic control, or merely dependent on the type of micro vessel.

The NF-κB signaling pathway plays a crucial role in regulating the expression of endothelial cell adhesion molecules such as E-selectin and VCAM-1 [24]. Several studies suggested that reduced Tie2 phosphorylation may lead to an increased expression of these adhesion molecules [11,12,13,25]. However, these studies did not report whether Tie2 was locally reduced in organs or whether reduced Tie2-phosphorylation levels coincided with changes in the cell adhesion molecule expression in particular microvascular segments in the organs. In our study, in renal arterioles of kidneys of mice with tamoxifen-induced reduced Tie2 levels, VCAM-1 protein expression was already high in quiescence compared to control groups and did not increase further upon LPS stimulation. In contrast, in alveolar capillaries with reduced Tie2 levels, E-selectin protein expression was increased upon LPS stimulation. Whether there is a direct relation between these observations and increased NF-κB activity, or whether other molecular mechanisms such as microRNA based posttranscriptional control of protein expression are responsible for this increase in the expression of cell adhesion molecules is not known at this time.

The protein expression and location of Tie2, E-selectin, and VCAM-1 were determined using immunohistochemistry. Disadvantages of this technique are the occurrence of day-to-day variation in staining intensity results, and difficult assessment of staining intensity by eye. To circumvent day-to-day variation, we stained each epitope in all mouse samples in one experiment. To make the intensity assessment objective, we used morphometric analysis of pixel counts. Western blot analysis or ELISA-based assessment of proteins could be considered for quantification purposes, yet both lead to loss of spatial information about the location of the proteins of interest. As our model induces heterogenous, though highly reproducible, Tie2 knockout which is dependent on the location of the endothelial cells in the organs, these other methods were not suitable for our aim. Furthermore, we focused, in this first study, on using the *Tie2^ΔE9^* model on a limited number of molecules that are related to acute inflammatory EC activation. More detailed analyses using, e.g., digital spatial profiling or laser microdissection of microvascular segments to create samples for further processing in (small) RNA sequencing workflow, may help to gain a broader view on the molecular changes that occur when specific endothelial cell subsets that lack Tie2 expression are exposed to pathophysiological conditions [26,27,28].

Aside from a highly heterogeneous distribution of Tie2 knockout in the mouse model employed, the extent of gene and protein expression was also highly variable between the individual mice in the experimental groups. This may partly be caused by variation in tissue sections and location in the organ in the case of immunohistochemical assessment of protein expression, yet significant variation also existed at the mRNA level between individual mice in arterioles in kidneys and capillaries in lungs. This variation is likely a combination of interindividual differences between mice exposed to sepsis-like conditions and endothelial cell heterogeneity within one microvascular segment [29,30]. How this heterogeneity and variation in response to LPS relates to organ function loss is unknown, though of importance for future translation of endothelial cell engagement in sepsis and their use as target cells for pharmacological intervention [31].

The 4 h time point was chosen as previous experiments showed that E-selectin and VCAM-1 expression induction occurred within the first 4–8 h after LPS exposure in this murine endotoxemia model, and did not further increase in time [19,32,33]. Furthermore, at this early time point, induction of E-selectin and VCAM expression was such that we would be able to detect either enhanced or diminished expression due to the absence of Tie2 in the organs studied. It is highly interesting to investigate, in future experiments, the effects of Tie2 knockout at later time points after initiation of sepsis in a chronic model such as the cecal ligation and puncture model, as it is suggested that patients suffering from sepsis experience conditions of prolonged low(er) endothelial Tie2 activation [34]. Understanding the broader consequences of the role of Tie2 in endothelial function and dysfunction in different microvascular beds in organs in time may serve drug development targeting the Tie2 system. In addition, it may also serve the accompanying development of biomarker panels that can identify spatiotemporal microvascular engagement in sepsis in a clinical setting [31].

Summarizing, our study confirmed that endothelial Tie2 is associated with endothelial cell responses to an inflammatory stimulus. Whether the role of Tie2 depends on microvascular bed (arteriolar versus capillary), or organ (kidney versus lung), or encompasses a combination of microvascular branch and organotypic control, remains to be established.

## Figures and Tables

**Figure 1 cells-12-01850-f001:**
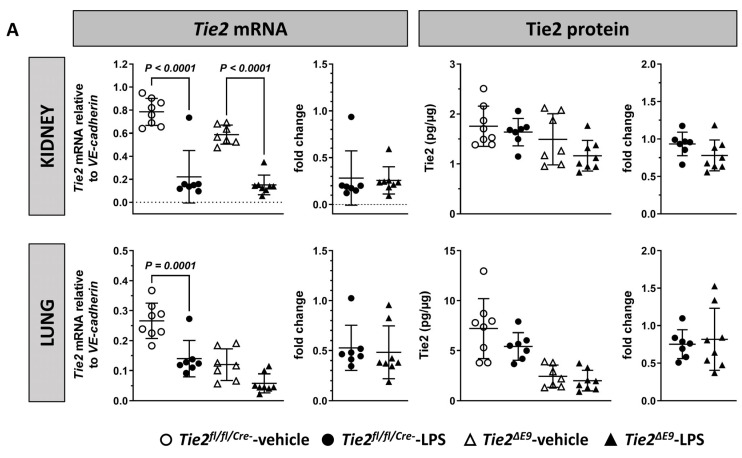
Effect of LPS-induced endotoxemia on Tie2 expression in kidneys and lungs. The expression of Tie2 mRNA and protein level was determined via RT-qPCR, and ELISA. (**A**) Graphs on the left side represent *Tie2* mRNA level, graphs on the right represent Tie2 protein levels. Graphs show individual values and means (black lines) ± SD (error bars). Fold change graphs represent mRNA, respectively protein levels in LPS-challenged groups normalized to levels in vehicle control groups. (**B**) Tie2 protein expression was localized using immunohistochemistry in kidneys and lungs of LPS-challenged *Tie2^fl/fl/Cre-^* control and *Tie2^ΔE9^* knockout mice. Photomicrographs of Tie2 staining, taken at 400× optical magnification, are shown for each genotype. (**C**) Positive pixels that represent Tie2 staining were quantified via morphometry of whole tissue sections (total) and of separate microvascular compartments within organs (kidney arterioles, and lung capillaries). Graphs represent % positive pixels in all groups (**left**), fold change graphs represent positive pixels in LPS-challenged groups normalized to positive pixels in vehicle control groups (fold change, **right**). Graphs show individual values and means (black lines) ± SD (error bars). Open circles: *Tie2^fl/fl/Cre-^*-veh mice (n = 8); closed circles: *Tie2^fl/fl/Cre-^*-LPS mice (n = 7); open triangles: *Tie2^ΔE9^*-veh mice (n = 7); closed triangles: *Tie2^ΔE9^*-LPS mice (n = 8). Differences in mRNA and protein levels between the four groups were calculated using one-way ANOVA with Sidak correction for multiple comparisons of selected pairs. Differences in fold change mRNA and protein levels between LPS-challenged *Tie2^fl/fl/Cre-^* control and *Tie2^ΔE9^* knockout mice were calculated using a two-tailed unpaired *t*-Test.

**Figure 2 cells-12-01850-f002:**
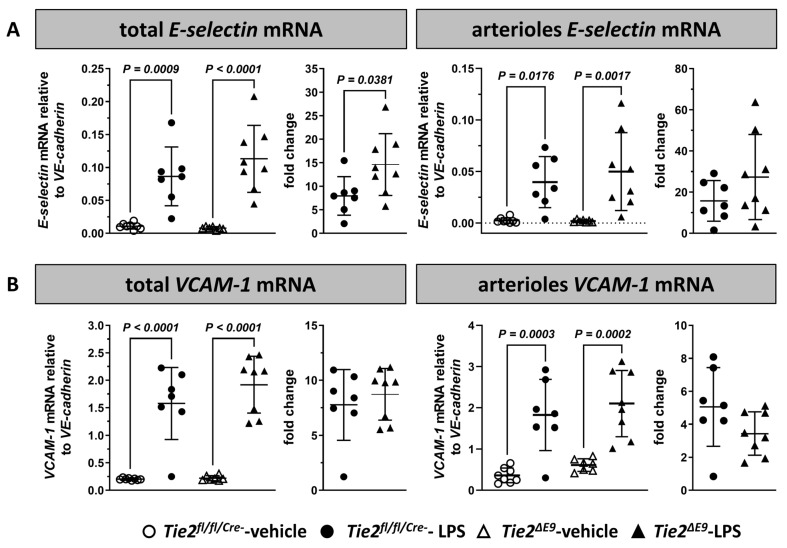
Effect of LPS-induced endotoxemia on the mRNA expression of inflammation-associated endothelial adhesion molecules in kidneys. The expression of mRNA was determined via RT-qPCR for (**A**) *E-selectin*, and (**B**) *VCAM-1*. mRNA expression levels were quantified in whole kidneys and in arterioles of kidneys of *Tie2^fl/fl/Cre-^* control and *Tie2^ΔE9^* knockout mice. Graphs represent mRNA levels in all groups (left), and mRNA levels in LPS-challenged groups normalized to levels in vehicle control groups (fold change; right). Graphs show individual values and means (black lines) ± SD (error bars). Basal mRNA expression levels of *E-*selectin and *VCAM-1* between genotypes did not significantly differ. Open circles: *Tie2^fl/fl/Cre-^*-veh mice (n = 8); closed circles: *Tie2^fl/fl/Cre-^*-LPS mice (n = 7); open triangles: *Tie2^ΔE9^*-veh mice (n = 7); closed triangles: *Tie2^ΔE9^*-LPS mice (n = 8). Differences in mRNA levels between the four groups were calculated using one-way ANOVA with Sidak correction for multiple comparisons of selected pairs. Differences in fold change mRNA levels between LPS-challenged *Tie2^fl/fl/Cre-^* control and *Tie2^ΔE9^* knockout mice were calculated using a two-tailed unpaired *t*-Test.

**Figure 3 cells-12-01850-f003:**
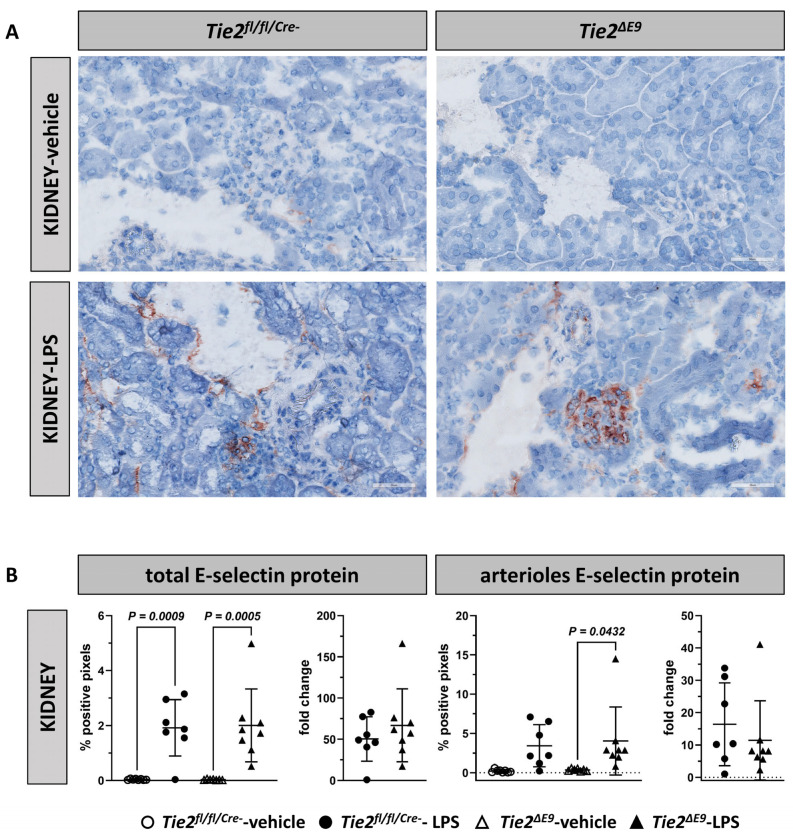
Location and morphometric analysis of inflammation-associated endothelial adhesion molecules in kidneys. The E-selectin and VCAM-1 protein expression was localized using immunohistochemistry. Photomicrographs of (**A**) E-selectin, and (**C**) VCAM-1 staining, taken at 400× optical magnification, are shown for each genotype. Positive pixels representing (**B**) E-selectin, and (**D**) VCAM-1 staining were quantified via morphometry of whole tissue sections (total), and of microvascular arterioles. Graphs represent % positive pixels in all groups (left), and positive pixels in LPS-challenged groups normalized to positive pixels in vehicle control groups (right; fold change). Graphs show individual values and means (black lines) ± SD (error bars). Open circles: *Tie2^fl/fl/Cre-^*-veh mice (n = 8); closed circles: *Tie2^fl/fl/Cre-^*-LPS mice (n = 7); open triangles: *Tie2^ΔE9^*-veh mice (n = 7); closed triangles: *Tie2^ΔE9^*-LPS mice (n = 8). Differences in protein levels between the four groups were calculated using one-way ANOVA with Sidak correction for multiple comparisons of selected pairs. Differences in fold change protein levels between LPS-challenged *Tie2^fl/fl/Cre-^* control and *Tie2^ΔE9^* knockout mice were calculated using a two-tailed unpaired *t*-Test.

**Figure 4 cells-12-01850-f004:**
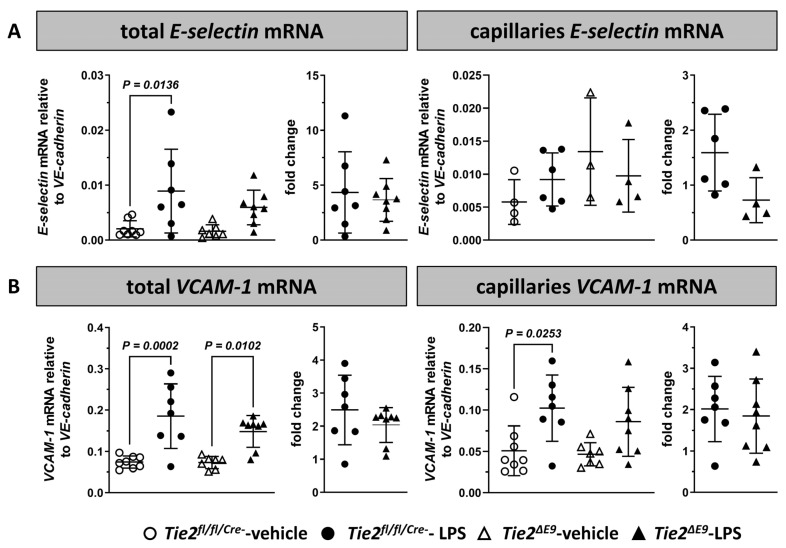
Effect of LPS-induced endotoxemia on the mRNA expression of inflammation-associated endothelial adhesion molecules in lungs. The expression of mRNA was determined via RT-qPCR for (**A**) *E-selectin* and (**B**) *VCAM-1*. mRNA expression levels were quantified in whole lungs and in alveolar capillaries of lungs of *Tie2^fl/fl/Cre-^* control and *Tie2^ΔE9^* knockout mice. Graphs represent mRNA levels in all groups (left), and mRNA levels in LPS-challenged groups normalized to levels in vehicle control groups (fold change; right). Graphs show individual values and means (black lines) ± SD (error bars). Basal mRNA expression levels of *E-*selectin and *VCAM-1* between genotypes did not significantly differ. Open circles: *Tie2^fl/fl/Cre-^*-veh mice (n = 8); closed circles: *Tie2^fl/fl/Cre-^*-LPS mice (n = 7); open triangles: *Tie2^ΔE9^*-veh mice (n = 7); closed triangles: *Tie2^ΔE9^*-LPS mice (n = 8). Differences in mRNA levels between the four groups were calculated using one-way ANOVA with Sidak correction for multiple comparisons of selected pairs. Differences in fold change mRNA levels between LPS-challenged *Tie2^fl/fl/Cre-^* control and *Tie2^ΔE9^* knockout mice were calculated using a two-tailed unpaired *t*-Test.

**Figure 5 cells-12-01850-f005:**
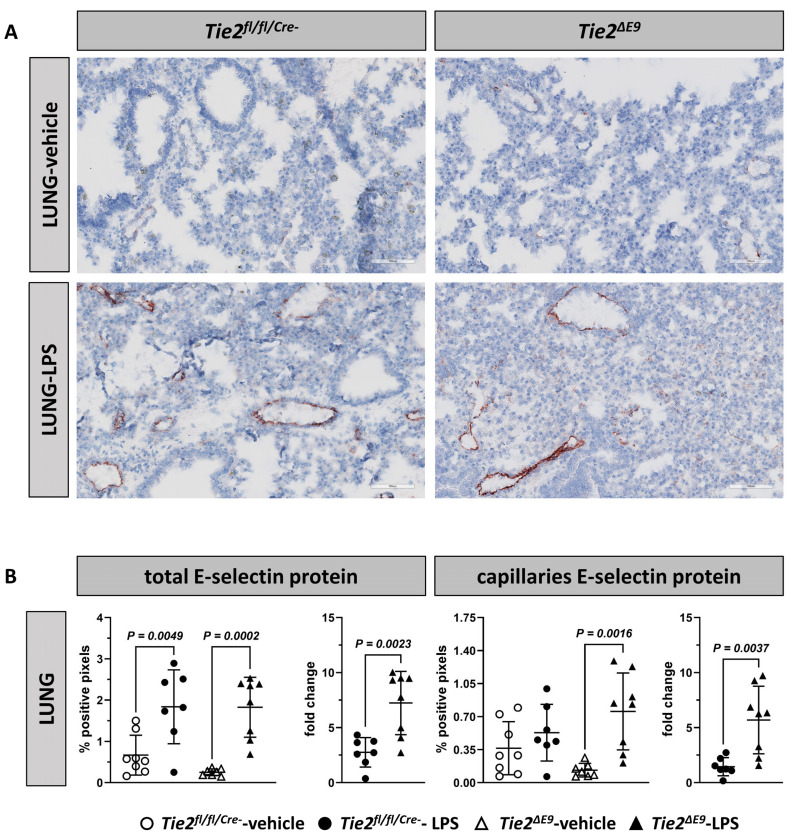
Location and morphometric analysis of inflammation-associated endothelial adhesion molecules in lungs. E-selectin and VCAM-1 protein expression was localized using immunohistochemistry. Photomicrographs of (**A**) E-selectin and (**C**) VCAM-1 staining, taken at 400× optical magnification, are shown for each genotype. Positive pixels representing (**B**) E-selectin and (**D**) VCAM-1 staining were quantified via morphometry of whole tissue sections (total) and of alveolar capillaries. Graphs represent % positive pixels in all groups (left), and positive pixels in LPS-challenged groups normalized to positive pixels in vehicle control groups (right; fold change). Graphs show individual values and means (black lines) ± SD (error bars). Open circles: *Tie2^fl/fl/Cre-^*-veh mice (n = 8); closed circles: *Tie2^fl/fl/Cre-^*-LPS mice (n = 7); open triangles: *Tie2^ΔE9^*-veh mice (n = 7); closed triangles: *Tie2^ΔE9^*-LPS mice (n = 8). Differences in protein levels between the four groups were calculated using one-way ANOVA with Sidak correction for multiple comparisons of selected pairs. Differences in fold change protein levels between LPS-challenged *Tie2^fl/fl/Cre-^* control and *Tie2^ΔE9^* knockout mice were calculated using a two-tailed unpaired *t*-Test.

**Table 1 cells-12-01850-t001:** RT-qPCR primers used in this study.

Gene	Assay ID	Encoded Protein
*Gapdh*	Mm99999915_g1	Glyceraldehyde-3-phosphate dehydrogenase (Gapdh)
*Tek*	Mm00443242_m1	Tyrosine kinase receptor (Tie2), CD202
*Cdh5*	Mm00486938_m1	Cadherin 5 (VE-cadherin)
*Esel*	Mm00441278_m1	Endothelial Leukocyte Adhesion Molecule 1, E-selectin
*VCAM-1*	Mm00449197_m1	Vascular Cell Adhesion Molecule 1

## Data Availability

The data presented in this study are published in the Figshare repository, accession number 10.6084/m9.figshare.23098880.

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
