# Peer review of "Reduced Tie2 in Microvascular Endothelial Cells Is Associated with Organ-Specific Adhesion Molecule Expression in Murine Health and Endotoxemia"

_cells, 2023, doi:10.3390/cells12141850_

Round 1

Reviewer 1 Report

The paper by Lucas et al, adresses the role of Tie2 in the response to LPS-triggered endotoxemia. This is a paper which could pave a way for further investigation using other (more appropriate) mouse models. However, the conclusions are well supported by the data. I have only some minor comments to make. 

Please include the information if the control group received tamoxifen (this compond has a well described impact on cardiovascular system.

Please specify if there were differences in the mean age between the inspected groups - the animals of significantly different age were used and it is not specified if both control and experimental group were derived from littermates.

For a better legibility of the graphs, it would be useful to include the legend - depicting which symbol corresponds to which group

Reviewer 2 Report

The manuscript entitled “Reduced Tie2 Expression in Renal Arterioles and Lung Capillaries is Associated with Organ Specific Reduction in Endothelial Adhesion Molecule Expression in Murine Endotoxemia” falls within the scope of the Journal. However, this reviewer has the following comments for the manuscript.

Major comments:

-   Authors report that “Additionally, in the presence of Angpt-2 increased expression of vascular cell adhesion molecule-1 (VCAM-1) and intercellular adhesion molecule-1 (ICAM-1) occurred in TNF-α-stimulated HUVEC, and Angpt-1 treatment of mice suppressed upregulation of E-selectin, VCAM-1 and ICAM-1 protein in lungs of mice exposed to LPS”. Do authors have data on the stimulation of HUVEC or HDBEC endothelial cells after stimulation with LPS? Authors should report their data and/or those found in the literature.

- Authors report that “The NF-κB signaling pathway plays a crucial role in regulating the expression of endothelial cell adhesion molecules such as E-selectin and VCAM-1 [23]”. Do authors have data ( by western blot?) on the expression of the NF-κB signaling pathway?

- Authors should report in each Figure legend the statistical analysis used.

-  Authors should insert a graphical abstract that summarizes the contents of the article in a concise form in order to capture the attention of the readership.

 Minor comments:

- Authors should report the keywords in alphabetical order.

- The English language has to be revised.

- Authors should improve the formal aspects of the manuscript.

-The English language has to be revised.

Reviewer 3 Report

Lucas et al. conducted an interesting study exploring the LPS-induced reactive expression levels of endothelial adhesion molecules in the Tie2 KO mice models. While providing valuable data, the manuscript has multiple concerns that need to be addressed before being considered for publishing.

  1. E-selection and VCAM-1 expression levels were evaluated 4 hours after endotoxemia started. So why do you choose 4 hours instead of other different time points?

  2. There are different turnover rates between mRNA and protein, so for the mRNA and protein expression levels of E-selection and VCAM-1, please add more data from other time points such as 8h, 12h, 24h,36h, or 48h.

  3. Based on result 3.1 and Figure 1, although there is a significant decrease in the nephric Tie2 mRNA level after LPS in both groups, the total and arterioles Tie2 protein levels remain similar. Please explain the possible reasons. 

Reviewer 4 Report

The involvement of endothelial cells in the microvasculature of organs as active contributors to the pathophysiology of sepsis is well recognized. While the precise underlying mechanism remains incompletely understood, the tyrosine-protein kinase receptor Tie2 has been implicated in the inflammatory response associated with sepsis. The authors' previous studies have successfully established a mouse model wherein Tie2 is specifically knocked out in endothelial cells, revealing a significant reduction in Tie2 protein levels in arterioles across various organs, as well as in the alveolar capillary microvascular bed within the lungs. Additionally, a recent study demonstrated that the partial deletion of Tie2 results in distinct microvascular endothelial responses to LPS that vary depending on the specific vascular bed and organ. In the present investigation, the authors further validated these findings by utilizing mice with an inducible deletion of Tie2 in endothelial cells, specifically focusing on the effects of LPS. Overall, the authors' work pertaining to the deletion of Tie2 in endothelial cells is commendable. However, it would be beneficial to highlight the novel discoveries and elaborate on their clinical implications. In order to enhance the manuscript's quality, please consider addressing the following concerns:

1.       Please revise the Abstract, focusing on highlighting all major findings and emphasizing their novelty, in addition to correcting any grammatical errors.

2.       Body weights were assessed during tamoxifen treatment to monitor the welfare of the mice. Please describe whether the administration of tamoxifen affects body weight gain.

3.       The pan-endothelial marker gene VE-cadherin was utilized as a reference gene to quantify Tie2 mRNA levels. Please justify its utilization considering that LPS may affect the gene expression of VE-cadherin.

4.       Please specify the reason why only the female mice were used.

5.       Due to the limited available data regarding the dynamics of Tie2 reduction over time, it is advisable to substitute the phrase "pre-existent reduced Tie2 expression" with, e.g., "tamoxifen-induced Tie2 reduction”.

6.       Please correct the errors in the references. For instance, citation of the reference 25 is incomplete.

7.       Please discuss the potential mechanisms and the clinical implications of microvascular bed-specific responses to LPS in the absence of Tie2 protein.

8.       Please revise this sentence, as it is confusing. “Our genetic mouse model was based on lacZ reporter studies showing that the stem cell leukemia (SCL) genetic driver exerted significant penetrance in small blood vessels in the majority of organs [17].”

9.       The manuscript contains numerous grammatical errors. Please seek editorial support.

The manuscript contains numerous grammatical errors. Please seek editorial support.

Round 2

Reviewer 2 Report

I have read the revised version of the manuscript named "Reduced Tie2 expression in renal arterioles and lung capillaries is associated with organ specific reduction in endothelial adhesion molecule expression in murine endotoxemia". The authors have made revisions to this article in accordance with the suggestions of reviewers. I think that manuscript is worth publishing in Cells.

Reviewer 3 Report

Accept

Reviewer 4 Report

The authors have adequately addressed my concerns. 

The reason I selected "Minor editing of English language required": please consider changing "organ specific"  into "organ-specific", in the tile. However, this might be able to be corrected in the production process.